# Regional multidisciplinary team intervention programme to improve colorectal cancer outcomes: study protocol for the Yorkshire Cancer Research Bowel Cancer Improvement Programme (YCR BCIP)

John Taylor [ID],[1] Penny Wright,[2] Hannah Rossington,[1] Jackie Mara,[1] Amy Glover,[3] Nick West,[3] Eva Morris,[1] Phillip Quirke,[3] On behalf of YCR BCIP study group

For numbered affiliations see end of article.

**Correspondence to**
Mr John Taylor;
j.c.taylor@leeds.ac.uk

## ABSTRACT

**Introduction** Although colorectal cancer outcomes in England are improving, they remain poorer than many comparable countries. Yorkshire Cancer Research has, therefore, established a Bowel Cancer Improvement Programme (YCR BCIP) to improve colorectal cancer outcomes within Yorkshire and Humber, a region representative of the nation. It aims to do this by quantifying variation in practice, engaging with the colorectal multidisciplinary teams (MDTs) to understand this and developing educational interventions to minimise it and improve outcomes.

**Methods and analysis** Initially, routine health datasets will be used to quantify variation in the demographics, management and outcomes of patients across the Yorkshire and Humber region and results presented to MDTs. The YCR BCIP is seeking to supplement these existing data with patient-reported health-related quality of life information (patient-reported outcome measures, PROMs) and tissue sample analysis. Specialty groups (surgery, radiology, pathology, clinical oncology, medical oncology, clinical nurse specialists and anaesthetics) have been established to provide oversight and direction for their clinical area within the programme, to review data and analysis and to develop appropriate educational initiatives.

**Ethics and dissemination** The YCR BCIP is aiming to address the variation in practice to significantly improve colorectal cancer outcomes across the Yorkshire and Humber region. PROMs and tissue sample collection and analysis will help to capture the information required to fully assess care in the region. Engagement of the region's MDTs with their data will lead to a range of educational initiatives, studies and clinical audits that aim to optimise practice across the region.

## Strengths and limitations of this study

► This study uses a novel approach by providing regional colorectal cancer multidisciplinary teams (MDTs) with data and, with their input, developing educational interventions to minimise any variation seen in order to optimise outcomes.

► The strength of this study includes evaluating variation in the management and outcomes of patients with colorectal cancer by combining routinely collected clinical datasets and novel information on aspects of care not currently quantifiable through existing datasets.

► The study will facilitate implementation of guidelines such as routine screening for Lynch syndrome and deficient mismatch repair status.

► The potential of the study is limited by the extent of engagement from regional colorectal cancer MDTs.

Although survival rates have improved over time, they continue to lag behind those attained by many economic neighbours[1] and so developing educational interventions to improve outcomes is a priority.

Some potential explanations for these survival differences include quality of care and access to services. For example, local recurrence and survival rates in rectal cancer are related to the quality of the surgical resection[2] so improving surgical quality is likely to improve outcomes. Likewise, ensuring gold-standard diagnostic, staging and multidisciplinary care across all aspects of the patient pathway have a demonstrable impact on outcomes[3] while screening for Lynch syndrome and deficient mismatch repair will enable more effective management of patients.

## INTRODUCTION

Colorectal cancer is a common disease in the UK with over 41 000 people diagnosed each year and 16 000 people dying from it.

Other countries have acknowledged these relationships and taken steps to improve standards. For instance, in the early 1990s, Norway recognised there was a huge variation in local recurrence rates depending on which surgeon operated.[4] In response, they initiated a training programme to quantify variation and improve the quality of surgery. This strategy was a success, reducing local recurrence and improving overall survival.[5] As a result similar programmes in other countries, including the UK, were established with equally positive outcomes including lower postoperative mortality[6] and better survival,[3 7] lower permanent stoma[7] and local recurrence rates,[8] better preoperative staging,[9] improved selection of patients for non-surgical treatment[9] and the need for less emergency surgery.[3]

While these programmes have undoubtedly had a positive impact, variations in management remain and particularly so in the UK.[10 11] Although survival rates in the UK have improved, the gap to rates attained in comparable countries[1] have not closed. A better understanding of what is driving these variations will help to target educational interventions to improve care.

Yorkshire Cancer Research (YCR) is a charity whose aim is to ensure that more people of Yorkshire and Humber 'avoid, survive and cope' with cancer and, by 2025, they intend to have reduced cancer deaths in the region by 2000 per year.[12] As part of their strategy to achieve this, they have recently funded the Bowel Cancer Improvement Programme (YCR BCIP). This initiative builds on the experience gained from English National programmes[9 13] and extends the approach of those successful programmes completed in Scandinavia.[3 5 8] It centres on the collection and analysis of robust colorectal cancer data to examine practice across the region, ensuring the multidisciplinary teams (MDTs) managing the disease engage with these data and agree on areas for improvement. Educational interventions or other strategies can then be developed and implemented to ensure optimal practice is achieved.

This protocol paper gives an overview the YCR BCIP study, which has the following aims:

1. Quantify and report the variation in the demographics, management and outcomes of the region's patients with colorectal cancer using
   a. Routinely collected clinical datasets.
   b. New information on aspects of care not currently quantifiable through existing datasets (eg, patient-reported outcome measures (PROMs), the molecular subtyping of the disease and the quality of radiology, pathology and surgery).
2. Determine how outcomes from the Yorkshire and Humber region compare to the rest of England.
3. Provide the Yorkshire and Humber region colorectal MDTs with these data and, with their input, develop educational interventions to minimise any variation seen in order to optimise outcomes.
4. Facilitate implementation of guidelines such as routine screening for Lynch syndrome and deficient mismatch repair status.
5. Evaluate improvement in outcomes over the study period.

## METHODS AND ANALYSIS

### Setting

The Yorkshire and Humber region of the UK has a population of approximately 5.7 million with around 3300 colorectal cancer diagnoses a year, amounting to just over 10% of diagnoses in England. Patients diagnosed with colorectal cancer in the region are managed by 16 different MDTs. These operate across 14 National Health Service (NHS) trusts, which provide health services for particular geographical areas of the region which exhibit a range of workloads, ranging between approximately 120 and 390 cases a year (4%–12% of the regional workload). Patients within the geographical areas covered by each trust exhibit a variable range of demographic features such as socioeconomic deprivation and comorbidity.[14]

### Programme overview

YCR BCIP will examine routinely collected NHS data and, subject to patient consent, additional data collected specifically for the programme. Baseline performance outcomes for each MDT from across the region will be assessed. The knowledge will be fed back to each MDT and a programme of educational interventions developed based on the performance outcomes. The process will be iterative as outlined in figure 1. The programme commenced on 1 April 2016 and will run until 31 March 2021.

Following a regional meeting of MDT representatives to discuss, plan and agree to programme participation held in June 2016, six specialty groups were initially established. The remit of these groups is to provide oversight and direction for their clinical specialty within the programme, to review data and analyses and to develop appropriate educational initiatives. The six groups focused on the main clinical disciplines involved in the MDT; surgery, radiology, pathology, clinical oncology, medical oncology and specialist nursing. Group membership was drawn from all of the region's MDTs and an individual identified from each to to act as a coordinator between the research team and MDT. The need for a separate anaesthetics workstream was subsequently identified and introduced. A lead clinician was also identified for each discipline, having the responsibility to coordinate educational events, gather opinion on best practice, formulate consensus views and drive agreed initiatives into routine clinical care across the region.

### Data sources

Although broad ranging, it was recognised that the routine colorectal cancer data available are not sufficient to quantify all aspects of care robustly. While routine data provide good information available on 'hard' outcomes such as which operation was undertaken or survival time, there is a lack of information on what this really means to the individual concerned and the quality of that survival.

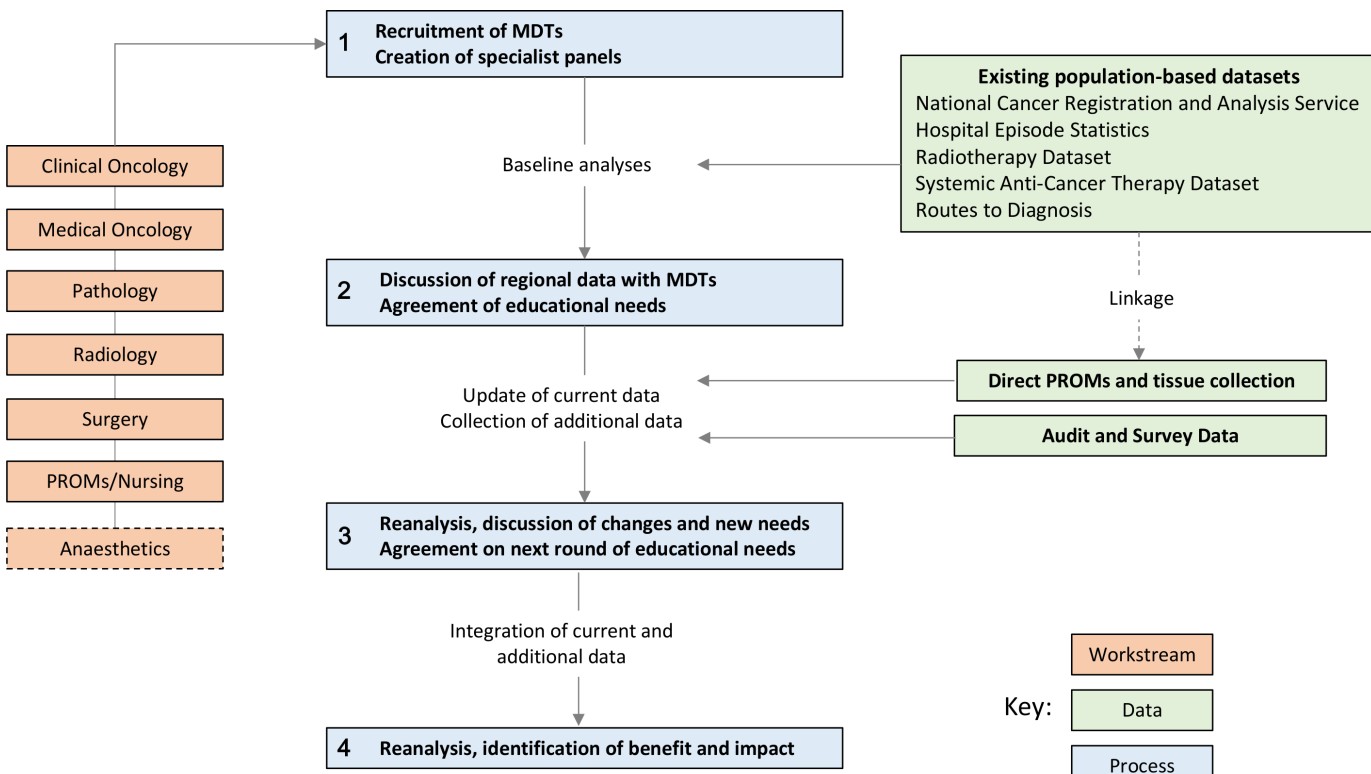

**Figure 1** Yorkshire Cancer Research Bowel Cancer Improvement Programme process and study design. MDT, multidisciplinary team; PROMs, patient-reported outcome measures.

This precludes the production of the rich intelligence the MDTs want to fully understand variation in care. Therefore, data collated in YCR BCIP will come from three main sources:

### Existing population-based datasets

Consisting of routine NHS datasets providing both regional data (around 3300 cases per year) and comparative national data (around 28 700 cases per year) on patient and tumour characteristics, treatment choices, diagnosis pathways and patient outcomes. All patients diagnosed with a first primary colorectal cancer (International Statistical Classification of Diseases and Related Health Problems 10th Revision (ICD-10) C18–C20) in England from 1 January 2005 to the end of the study and registered in the National Cancer Registration and Analysis Service (NCRAS)[15] will be eligible.

### Direct PROMs and tissue collection

Consisting of regional PROMs data collected directly from patients via a questionnaire on health-related quality of life (at both the time of diagnosis, before primary treatment if possible and again at 12 months postdiagnosis) and molecular testing of tumour and tumour-associated normal mucosal tissue samples.

All MDTs in the region have been invited to participate in this element of the YCR BCIP. Every patients with colorectal cancer (ICD-10 C18–C20) are eligible if they are considered suitable for treatment, English language literate, aged at least 18 years old and with capacity to give informed consent. Participants will be asked to consent to the study prior to commencement of primary treatment or following an emergency intervention.

### Audit and survey data

Consisting of unlinked anonymised data collected from clinicians at regional MDTs in the form of an audit or a clinician survey. The exact nature of these shall be identified by each clinical discipline depending on the needs and availability of existing data. However, they are expected to include but are not limited to: an audit on surgical quality using specimen photographs, audits assessing the completeness in the recording of pathology and radiology reports, an audit on the methods of lymph node retrieval and clinican surveys assessing the management of patients in the oncological, surgical and aesthetics settings. All clinicians who are members of regional MDTs as part of the discipline being assessed will be eligible to participate.

### Data collection
#### Existing population-based datasets

The YCR BCIP seeks to make maximum use of routine NHS datasets and these are brought together in the UK Colorectal Cancer Intelligence Hub[16] where data from the NCRAS are linked to other datasets relevant to colorectal cancer to provide the richest data possible and enable analysis of the full cancer pathway. These include, but are not limited to: Hospital Episode Statistics (HES), Radiotherapy Dataset, Systematic Anti-Cancer Therapy Datatset

and Routes to Diagnosis. To provide a baseline, these data will cover the period from 1 January 2005 to the start of the programme and be routinely collected until the end of the study to assess changes throughout it.

All patients in the NCRAS data will be assigned an MDT using the HES procedure closest to the patient's diagnosis date. If no procedure is found, the closest inpatient or outpatient appointment to the diagnosis date at a hospital with a colorectal MDT is used. Those not assigned an MDT (<1% of patients) will be excluded from analyses. The assigned MDT will be assumed to have been responsible for the patient's management and treatment options.

### Direct PROMs and tissue collection

This research project is adopted by the NIHR Yorkshire and Humber Clinical Research Network and therefore recruitment will be undertaken across the region by network research and clinical staff working collaboratively. Recruitment will run over a 30-month period.

Eligible patients will be identified and approached in two ways.

1. Identified via the MDT and informed about the study by consultant letter sent out with their appointment letter for the primary preassessment clinic visit . Where possible, patients will be approached about the study at this clinic appointment. Patients missed at this appointment will be contacted at the earliest convenient time point and informed about the study.
2. Identified by their NHS clinical team if they present as an acute admission (eg, with a bowel obstruction) and informed about the study by their clinical team following the emergency intervention (eg, surgery).

Participants will have the option of completing a PROMs questionnaire either online PROMs using the University of Leeds secure questionnaire administration system QTool,[17] accessed via the study website, (www.YCRBCIP. leeds.ac.uk) or on paper PROMs with a prepaid return envelope. They will be asked to complete the questionnaire as soon as possible after consenting. This may be completed while attending hospital at the time of consent or later at home. Just prior to the 12-month follow-up questionnaire, the patient status will be checked via NCRAS to confirm that the patient is still alive. Patients will then be sent an email/letter 12 months postdiagnosis by the YCR BCIP research team, inviting them to complete PROMs again (on paper or online according to patient preference). At both time points, reminder letters or emails will be sent at 2 weeks and followed up 2 weeks later (if no response) with the questionnaire being resent with the reminder letter. The process is outlined in figure 2.

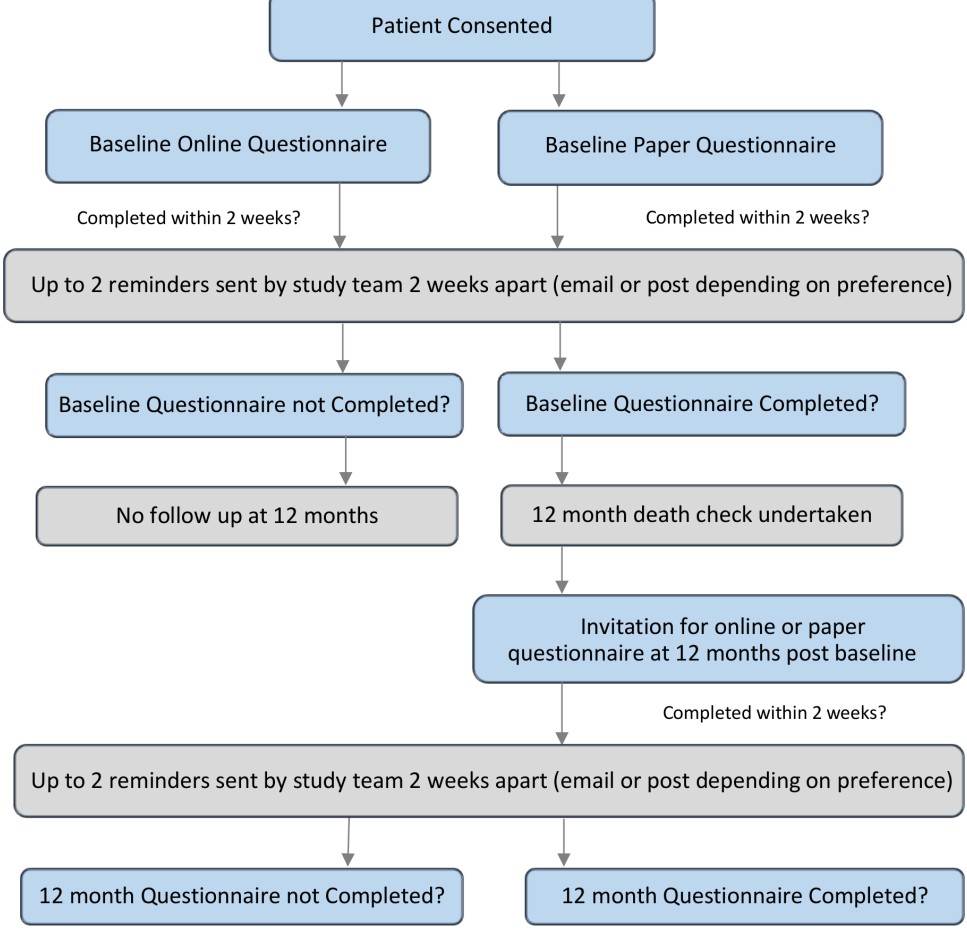

**Figure 2** Process for the patient-reported outcomes measures study.

Participants will be asked to complete a range of validated generic, cancer-specific and colorectal cancer-specific PROMs in addition to sociodemographic characteristics. Content has been informed by: clinical relevance, opinion of service users, overall length and participant burden,[18] reviews of questionnaires measuring quality of life in patients with colorectal cancer[19–23] and UK recommendations for the core outcomes set for trials in colorectal cancer surgery.[24]

The questionnaire comprises four sections at baseline and five sections at follow-up. The questionnaire is estimated to take 30–35 min to complete.

### Your overall health and quality of life (both time points)
► European Organisation for Research and Treatment of Cancer (EORTC) Quality of Life Questionnaire-C30, the colorectal cancer-specific module and items from the prostate, endometrial and cervical cancer modules.[25–29]
► EuroQol-Five Dimension-Five Level Group.[30]

### Your everyday life and well-being (both time points)
► Social Difficulties Inventory-21.[31 32]
► The Short Warwick-Edinburgh Mental Well-being Scale.[33 34]

### Managing your health (both time points)
► Self-efficacy for Managing Chronic Disease.[35]
► The Brief Illness Perceptions Questionnaire.[36]

### Questions about you (both time points)
► Self-reported sociodemographic and clinical details.

### The financial cost of cancer (second time point only)
► A questionnaire developed in-house based on one used in the electronic Patient-reported Outcomes from Cancer Survivors (ePOCS) study.[37]

The tissue samples to be collected for this study are:
1. Excess pretreatment, diagnostic biopsy tumour tissue.
2. Excess tissue following surgical resection (tumour and tumour-associated normal mucosal tissue).
3. Excess tumour tissue following the biopsy or resection of distant metastases.

The tissue samples will be formalin fixed and paraffin wax embedded as per the routine histopathology procedures at the patients' hospital. Tissue blocks will be used for any routine clinical diagnostic procedures required by the hospital before being sent to Pathology at the University of Leeds, and will be available to return to the hospital for further clinical testing if required.

Based on the number of diagnoses per year, there is potential to consent approximately 8250 patients from sites across the region over a 30-month recruitment period. However, due to eligibility restrictions, consent by MDT and by individuals, administrative/practical issues and staggered site recruitment start dates, the participation rate is likely to be less. Using estimates based on previous studies,[38 39] the consent rate is estimated to be around 42% and the an attrition rate of approximately

35% at the 12 months follow-up point, giving a sample size of at least 1200. This will provide MDTs with a comprehensive descriptive profile of their patients in terms of quality of life outcomes; more complex analyses will be undertaken if sufficient numbers are accrued.

### Audit and survey data
Audit data will be limited to data that are collected as part of routine patient care. For example, anonymised MRI scans that have been performed as a part of patient care may be used as part of an educational training initiative. No specific recruitment will be performed for these data as only anonymised data used as part of routine patient care will be used. Clinicians at regional MDTs will complete surveys anonymously online using www.online-surveys.ac.uk (formally Bristol Online Survey). These surveys will be used to assess clinical practice and patient management. For example, a survey regarding the use of adjuvant chemotherapy will be used to compare regional practice to the most recent evidence basis and used to inform a regional guideline for the treatment of these patients.

Regional MDT clinicians will be invited to complete the online surveys via email and through specialty group meetings.

### Consent
#### Existing population-based datasets
The data are derived from the patient-level information collected by the NHS, as part of the care and support of patients with cancer. NCRAS has a specific legal permission to collect this information without the need to seek consent, however, patients can ask NCRAS to remove their details from the cancer registry at any time. Access to cancer registration data and the other routine health datasets used in this study is controlled by the Public Health England (PHE) Office for Data Release, and is only approved for permitted medical purposes.[15] This work is covered by a data sharing contract with PHE (ODR1516_369).

#### Direct PROMs and tissue collection
Identified patients will be provided with full study information (written and verbal) by specialist research nurses or Clinical Nurse Specialists who have undergone Good Clinical Practice training to assure the rights, safety and well-being of research participants are protected.[40] Written consent may be taken at the time of this approach but patients will be given up to a week to think about study participation. Patients who wish to join the study will be asked to read, complete and sign a consent form, including their contact details name, address and/or email address. The person taking consent will also record the patient's date of birth, NHS number and gender. Patients must consent to both PROMs participation and tissue collection to be included in the study.

Patients are asked to consent to their clinical team being informed in the event that clinically relevant laboratory

results are found. This includes the possibility of hereditary conditions identified through tumour testing, 'for example, Lynch syndrome or deficient mismatch repair status'. These will be confirmed through routine NHS Clinical Genetics testing after counselling following referral from the local clinical teams with no germline testing taking place through the programme.

### Audit and survey data

No specific consent will be needed for these data as only anonymised data used as part of routine patient care will be used, for example, specimen photographs or scans will have all identifiers removed. Consent for the surveys will be implied when the clinician completes a survey that they have been invited to.

### Data linkage

While initially the existing population datasets and direct PROMs and tissue collection will be analysed separately, these will subsequently be linked together through the UK Colorectal Cancer Intelligence Hub using name, NHS number and date of birth. This will provide additional patient characteristics for analysis of the PROMs and tissue data.

### Analysis

#### Existing population-based datasets

Baseline assessments of care in the region are to be performed on individuals diagnosed with colorectal cancer in Yorkshire and Humber, enabling comparison between teams in the region and with national data.[14] Initially, this includes using descriptive analysis and statistical methods such as regression modelling, survival analysis and funnel plots[41] comparing the following data: demographic characteristics, tumour characteristics, surgery and oncology management and short and long-term outcomes.

Some analyses will be rerun periodically over the course of the programme to evaluate the impact on outcomes of specific educational interventions. The measures to be analysed and the sources of these can be found in table 1.

#### Direct PROMs and tissue collection

Descriptive statistics will be used to report the questionnaire results and assess the quality of life outcomes of the participants. Following data linkage, the outcomes will be analysed according to stage of disease, treatment type, comorbidity, age, ethnic and sociodemographic group (and other relevant variables). These descriptive analyses will identify potential relationships of interest, which can be investigated further. Regression modelling will be used to investigate associations among the different types of variables to identify statistically and clinically significant risk factors and predictors of outcomes. In order to be robust, analyses will require appropriate adjustment for casemix and other confounding factors and may require more complex techniques, such as the modelling of hierarchies within the data (multilevel modelling), post hoc

**Table 1** Measures of patient characteristics, treatment measures and outcomes to analysed and the corresponding data source

| | Data source | | |
| --- | --- | --- | --- |
| | Existing population-based datasets | Direct PROMs and tissue collection | Audit and survey data |
| **Patient and tumour characteristics** | | | |
| Age and sex | NCRAS | PROMs | |
| Ethnicity | NCRAS | PROMs | |
| Height and weight | | PROMs | |
| Comorbidity | NCRAS | PROMs | |
| Socioeconomic status | NCRAS | PROMs | |
| Stage and site | NCRAS | | |
| Method of admission | NCRAS | | |
| **Treatment variation** | | | |
| Surgical resection rate | HES | | |
| Quality of surgery | | | Audit |
| Abdominoperineal excision rate | HES | | |
| Use of adjuvant and palliative chemotherapy | SACT | | Survey |
| Use of neoadjuvant radiotherapy | RTDS | | |
| Use of laparoscopic surgery | HES | | |
| Emergency care procedures | HES | | Survey |
| Practice of anaesthetics | | | Survey |
| Quality of MRI reporting | | | Audit |
| Quality of CT imaging | | | Audit |
| Liver metastases resection rate | HES | | |
| Nodal yields and retrieval methods | NCRAS | | Audit |
| **Outcomes** | | | |
| 30-day postoperative mortality | NCRAS | | |
| 1–5 years overall and net survival | NCRAS | | |
| 18-month postoperative stoma rate | HES | | |
| Postoperative hospital stay | HES | | |
| Emergency readmission rates | HES | | |
| Overall health and quality of life | | PROMs | |
| Everyday life and well-being | | PROMs | |
| Self-efficacy for managing chronic disease | | PROMs | |
| Financial cost of cancer | | PROMs | |

Continued

**Table 1** Continued

| | Data source | | |
| --- | --- | --- | --- |
| | Existing population-based datasets | Direct PROMs and tissue collection | Audit and survey data |
| Urinary function and faecal incontinence | | PROMs | |
| Sexual functioning | | PROMs | |
| Lower anterior resection syndrome | | PROMs | |
| Molecular subtyping | | Tissue | |

HES, Hospital Episode Statistics; NCRAS, National Cancer Registration and Analysis Service; PROMs, patient-reported outcome measures; RTDS, radiotherapy dataset; SACT, systemic anticancer therapy dataset.

weighting to overcome response bias and multiple imputation of missing data.

Tissue samples removed at surgery or biopsy which are surplus to routine clinical requirements will be used by the research team for upfront testing of novel biomarkers. YCR BCIP will test for novel or rare potentially treatable targets via a range of molecular techniques. The immunohistochemical markers of interest include, HER2, PTEN, PD-1 and PD-L1, amphiregulin and epiregulin, and immune system markers CD3 and CD8. The study will also undertake phenotype analysis by using high-resolution scanned images of tumours using novel algorithms to identify improved prognostic and predictive markers of outcome. Next-generation sequencing and/or pyrosequencing will be performed on any extracted DNA to identify tumours with specific gene hotspot mutations including, KRAS, NRAS, BRAF, EGFR and PIK3CA. Other molecular markers linked to causation, outcome and response to therapy will also be investigated, for example, bacterial toxin carriage or fusobacterium nucleatum. Analysis of RNA expression levels of receptor ligands, such as amphiregulin and epiregulin, and HER2 and HER3, may also be performed. The biomarkers will be linked to patient data (population based and PROMs) and undergo regression modelling to identify associations to understand how tumour biology influences outcomes, response and quality of life and how tumour biology can be influenced by lifestyle.

## Audit and survey data
The analyses of audit and survey data will be dependent clinical specialties involved and the nature of the data collected. For example, the completion of MRI scans for rectal cancer at an educational training initiative would be assessed by agreement coefficients and results of surveys will be analysed using descriptive statistics.

## Data safeguards
Participant recruitment will be undertaken by each trust involved in the study. Each research network site will allocate a study ID for potential participants. They will use a University of Leeds secure electronic transfer system every 2 weeks to inform the YCR BCIP research team of all recruitment activity. This will include the consented patients contact details, date of birth and NHS number to allow for follow-up and to ensure tissue blocks are appropriately labelled for tracking and data linkage. All subsequent participant contact will be undertaken by the central YCR BCIP team.

Storage of all hard copy documents will be in locked metal filing cabinets in research offices of the University of Leeds with secure access building controls. Tissue samples will be used and stored in a secure building with restricted access. Electronic data with pseudonymised (allocated ID number) patient information will be stored in a secure environment. These files will only be accessible to relevant members of the study analysis team. Where temporary storage of sensitive data is required (eg, contact details for sending out repeat questionnaires), files will be accessible only to relevant members of the research team and not stored with any linked data. Members of the analysis team will not have access to any identifiable data. Hard copy data will be kept for 5 years following the end of the study (until 2026) for long-term follow-up.

## Development of educational interventions
The analyses will be disseminated and discussed by regional MDTs at events for each clinical discipline. Agreements of educational needs to improve care will be agreed along with any additional data capture and audit processes that teams agree are required. The process is repeated with ongoing data analyses to establish the improvements in management and outcomes that have occurred (figure 1). As YCR BCIP progresses, further work will be undertaken looking at other outcomes including, but not limited to, those related to screening, pathology and long-term outcomes.

## Public patient involvement
Patients and carers were actively engaged through the PROMS working group to develop the design and content of the patient questionnaires, the patient information sheet and consent form. At the request of the patients and carers, additional questions were included around the financial impact of cancer and a specific request was made to EORTC to amend the EORTC colorectal module and add specific questions from other EORTC modules to understand side effects of cancer and cancer treatments. EORTC granted these specific amendments for this study. The electronic and paper copies of the final draft questionnaires were tested with patients attending a colorectal cancer follow-up clinic at one of the region hospitals. Modification to the layout of the questionnaires were made following the results of the testing. The testing gave an understanding of the length of time it took patients to complete the questionnaires. The PROMS working group will remain active throughout the length of the study; the

group will be kept apprised of recruitment levels and early results. It is expected that the patients will advise on the analysis and how the results are communicated to the regional clinical teams and wider audiences.

## DISCUSSION

The YCR BCIP study ambitiously aims to improve outcomes for patients with colorectal cancer across the Yorkshire and Humber region by in-depth analysis of existing and newly captured data, and by actively engaging local MDTs. In its initial stages, this will be done by demonstrating the variation in the demographics, management and outcomes in the region using routine NHS datasets. However, given the limitations of what can be achieved with existing data, the YCR BCIP is collecting additional data to analyse alongside this with the purpose to better understand what is driving the observed variation.

The PROMs data will enrich other study data and allow for an 'in-depth' description of what life is really like for patients with colorectal cancer at diagnosis and a year later. At present, the patient voice is not 'measured' routinely as part of clinical practice, neglecting the impact of illness and treatment on the everyday lives of patients. YCR BCIP will change this with the integration of PROMs administration into clinical practice.[39] The PROMs used in this study have been selected with input from experts in colorectal cancer and psychosocial care: patients, clinical nurse specialists and doctors. Although the length of the questionnaire would not be feasible to administer in everyday practice, information on which PROMs provide the most meaningful data will be obtained. In the future, risk-stratified follow-up may incorporate clinical indicators and key quality of life indicators to inform the best supportive care.[42]

Results from the tissue collection and testing could impact on treatment and follow-up decisions for the participating patients and, potentially, their families. For example, the results may indicate that a patient could benefit from a targeted treatment being tested through an open clinical trial if they develop an indication for further treatment, for example, Medical Research Council FOCUS4.[43] Increased risks of having a hereditary condition may be identified, which has implications for the patient and their family.

The YCR BCIP is offering a novel approach to address the variation in colorectal cancer care across a large region of the UK. Engagement of the region's MDTs with their data will lead to a range of educational initiatives, studies and clinical audits that aim to optimise practice across the region. It is planned that the outcomes of these will be presented to the relevant specialty group for review and to develop actions based on findings. The main limiting factor for the success of the study is that to understand the overall picture of colorectal cancer care and the ability to improve this in the region, relies on the extent of engagement from MDTs.

**Author affiliations**
¹Section of Epidemiology and Biostatistics, University of Leeds, Leeds Institute of Cancer and Pathology, Leeds, UK
²Section of Patient Centred Outcome Research, University of Leeds, Leeds Institute of Cancer and Pathology, Leeds, UK
³Section of Pathology and Tumour Biology, University of Leeds, Leeds Institute of Cancer and Pathology, Leeds, UK

**Acknowledgements** The YCR BCIP study group includes PQ, Paul Finan, PW, Nicholas West, Matthew Seymour, EM, David Sebag-Montefoire, Daniel Swinson, Damian Tolan, Simon Howell, Peter Brown, JT, AG, Aidan Hindley, HR, JM and Emily Boldison. The YCR BCIP research team acknowledges the support of the National Institute for Health Research Clinical Research Network (NIHR CRN).

**Collaborators** YCR BCIP study group: Paul Finan, Nicholas West, Matthew Seymour, David Sebag-Montefoire, Daniel Swinson, Damian Tolan, Simon Howell, Peter Brown, Aidan Hindley, Emily Boldison.

**Contributors** PQ is the principal investigator. NW, PW and EM are coinvestigators and/or workstream leads. Together these authors conceived and designed the study. HR and JM managed the study. JT provided statistical support. AG provided pathology support. All authors contributed to writing of the manuscript and have approved a final version.

**Funding** This work was funded by Yorkshire Cancer Research (Award reference number L394). PQ holds an NIHR Senior Investigator award. The UK Colorectal Cancer Intelligence Hub is supported by Cancer Research UK (C23434/A23706).

**Competing interests** None declared.

**Patient consent for publication** Not required.

**Provenance and peer review** Not commissioned; externally peer reviewed.

**ORCID iD**
John Taylor http://orcid.org/0000-0002-2518-5799

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
