## [Reviewer comments · BMJ Open]

ARTICLE DETAILS

TITLE (PROVISIONAL)	A regional multidisciplinary team intervention programme to improve colorectal cancer outcomes: study protocol for the Yorkshire Cancer Research Bowel Cancer Improvement Programme (YCR BCIP)
AUTHORS	Taylor, John; Wright, Penny; Rossington, Hannah; Mara, Jackie; Glover, Amy; West, Nick; Morris, Eva; Quirke, Phillip

VERSION 1 – REVIEW

REVIEWER	Cynthia Kendell, Research Associate Department of Surgery, Nova Scotia Health Authority, Canada
REVIEW RETURNED	08-May-2019

GENERAL COMMENTS	In their manuscript, entitled “The Yorkshire Cancer Research Bowel Cancer Improvement Programme (YCR BCIP): study protocol”, the authors have proposed the creation of a linked dataset, consisting of routinely collected data, PROMs, and tissue samples. These data will be used to study demographics, clinical care, and patient outcomes for individuals diagnosed with colorectal cancer in one geographic region and to inform the development of interventions to improve outcomes. While this dataset has the potential to be a powerful research platform, the manuscript requires substantial revisions to improve clarity and completeness. Please see comments below. General comments/edits: • Does the proposed work involve a data linkage? Will PROMs and tissue samples be linked to the NHS datasets? There is mention of data linkage under “Management of consent and follow-up procedures”, and later of data being “brought together through the UK Bowel Cancer Intelligence Hub”, but no clear explanation of the linkage process.• Data Safeguards-There is a large amount of data being collected about individual patients in this study. This raises important privacy concerns, particularly since it includes biological samples which are inherently identifiable, and especially if there is a data linkage. The authors should describe in detail the safeguards that will be undertaken to protect this dataset and to maintain patient privacy and confidentiality. For example, will direct identifiers be removed? Where will the dataset be housed? How will it be accessed and by whom? This may warrant a separate “Data Safeguards” section.• There is a lot of information provided in the manuscript; however, it is difficult to get a clear sense of how the process will unfold. The sections/subsections that are currently in use do not always flow well and often contain content that is redundant or would be best suited under a different heading. To improve flow
--

	and clarity, as well as completeness, consider using the following headings:  o Introduction o Methods and Analysis [ ] Setting [ ] Program Overview—The oversight of clinical specialties can be included here here. [ ] Study Cohort/Participants—Who will be involved? Both in terms of “data subjects” from the NHS datasets, and individuals who will participate directly. [ ] Cohort Identification/Participant Recruitment [ ] Consent—In addition to explaining consent process for PROMs and tissue samples, explain consent requirements for use of NHS datasets (e.g., whether consent is required and/or a waiver of consent will be obtained). [ ] Data Collection—It may be helpful to frame data collection in terms of data collected from existing databases, and data collected directly from study participants (rather than categorizing as consent vs. no consent). [ ] Data Linkage [ ] Data Analysis [ ] Data Safeguards [ ] Development of Educational Interventions o Discussion o Ethics and Dissemination Specific comments/edits:  • Throughout manuscript:  o In some places “interventions” are mentioned, in others “educational interventions”. Will interventions be limited to educational interventions or might other interventions also be developed? Please clarify. • Strengths and Limitations (bullets):  o Specify that the points listed are strengths/novel contributions. • Introduction  o Paragraph 3, sentence 2—Please provide the original citations(s) for work(s) showing that outcomes varied by surgeon. o Paragraph 3, sentence 3—Consider moving the citation for Wiebe et al 2002 to the end of sentence 3. o Paragraph 3, sentence 6—The sentence states that the implementation of programs was associated with improvements in 7 different outcomes. Were there improvements in all 7 outcomes for each of the programs/initiatives cited? If not, use the citations to indicate which program was associated with improvements in each specific outcome. o Paragraph 4, sentence 2—This sentence states “although outcomes have improved”. What outcomes are being referred to? The term “outcomes” is very broad—are these surgical outcomes, long-term outcomes such as overall or disease-free survival? Please be explicit and support with references where possible. o Paragraph 5, sentence 4—Consider “examine practice” rather than “quantify practice”. Also, change from “agree areas” to “agree on areas”. o Paragraph 6, sentence 1—Should “YCRBCIP” be “YCR BCIP” as it appears earlier on? o Paragraph 6, #2—This is the first time Humber is mentioned. Ensure consistency of use throughout. • Methods and Analysis  o Data Collection
--	---

	 [ ] It would be helpful if information on all of the individual data sources were addressed under the heading of Data Collection. [ ] 2(c) is "Screening for Lynch syndrome and deficient mismatch repair on routinely diagnosed bowel cancers". How is this a data source? Will all of these individuals be screened as part of the study? Or is screening data being collected from elsewhere? Please clarify. [ ] Will there be a patient-level linkage between the various data sources? If so, please provide details (preferably in a separate section focusing entirely on linkage).  [ ] Patient Recruitment  [ ] Who exactly will be approaching patients and inviting them to participate/have their data included in the study? If a member of patient's clinical care team, this raises issues around coercion. How will this be addressed? [ ] Patient Participation [ ] The majority of content in this section is related to PROMs and could be described under a description of PROMs (ideally under "Data Collection").  [ ] Analysis  [ ] I suggest that the authors list and define all individual variables that will be included in the study, and identify the relevant data source(s) for each. This can be done in a table. [ ] The manuscript details various data sources, but they do not all appear in the analysis section. As such, it is unclear how all of the data sources will be used. This section would benefit from additional information. For example, what are the specific research questions that the authors plan to address first? Are there specific planned sub-studies? Will there be a multi-phased approach? [ ] References  [ ] #4 and #5 refer to the same article. Please address error and update list accordingly.
--	---

REVIEWER	Paul Ritvo PhD York University, Toronto, Canada
REVIEW RETURNED	13-Jun-2019

GENERAL COMMENTS	This protocol paper write-up is deceptively simple as the project has required a great deal of multidisciplinary coordination to come about. I can appreciate the emphasis on streamlining the complexities involved but I believe the authors have gone too far in this regard. I would like to read more background on how the necessary coordination was accomplished. I also think that more detail is required on the 'routinely collected clinical datasets' re: what exactly they are and how they are routinely collected. With respect to the 'new information on aspects of care not currently quantifiable through existing datasets' how are these new data going to be collected, i.e. what deviations from standard procedures are being enacted and how? We need to know these additional details to evaluate quality and to possibly emulate the effort in other jurisdictions.
--

VERSION 1 – AUTHOR RESPONSE

REVIEWER FEEDBACK

In their manuscript, entitled “The Yorkshire Cancer Research Bowel Cancer Improvement Programme (YCR BCIP): study protocol”, the authors have proposed the creation of a linked dataset, consisting of routinely collected data, PROMs, and tissue samples. These data will be used to study demographics, clinical care, and patient outcomes for individuals diagnosed with colorectal cancer in one geographic region and to inform the development of interventions to improve outcomes. While this dataset has the potential to be a powerful research platform, the manuscript requires substantial revisions to improve clarity and completeness. Please see comments below.

General comments/edits:

- Does the proposed work involve a data linkage? Will PROMs and tissue samples be linked to the NHS datasets? There is mention of data linkage under “Management of consent and follow-up procedures”, and later of data being “brought together through the UK Bowel Cancer Intelligence Hub”, but no clear explanation of the linkage process.

Although the population based data will be analysed separately for the baseline analyses, the PROMs and tissue data will then be linked to the routine datasets to allow for more detailed analysis. We have tried to make this clearer in the manuscript.

- Data Safeguards-There is a large amount of data being collected about individual patients in this study. This raises important privacy concerns, particularly since it includes biological samples which are inherently identifiable, and especially if there is a data linkage. The authors should describe in detail the safeguards that will be undertaken to protect this dataset and to maintain patient privacy and confidentiality. For example, will direct identifiers be removed? Where will the dataset be housed? How will it be accessed and by whom? This may warrant a separate “Data Safeguards” section. This is a good point and the study does have extensive safeguards - which now has a separate section.

- There is a lot of information provided in the manuscript; however, it is difficult to get a clear sense of how the process will unfold. The sections/subsections that are currently in use do not always flow well and often contain content that is redundant or would be best suited under a different heading. To improve flow and clarity, as well as completeness, consider using the following headings: We agree that there is a lot of information and have taken on board the suggested layout below, with subsections for each data source where necessary. We have also adjusted the flow diagram in Figure 1 to related better to this layout.

- o Introduction
- o Methods and Analysis
- o Setting
- o Program Overview—The oversight of clinical specialties can be included here
- o here.
- o Study Cohort/Participants—Who will be involved? Both in terms of “data subjects” from the NHS datasets, and individuals who will participate directly.
- o Cohort Identification/Participant Recruitment
- o Consent—In addition to explaining consent process for PROMs and tissue samples, explain consent requirements for use of NHS datasets (e.g., whether consent is required and/or a waiver of consent will be obtained).
- o Data Collection—It may be helpful to frame data collection in terms of data collected from existing databases, and data collected directly from study participants (rather than categorizing as consent vs. no consent).
- o Data Linkage
- o Data Analysis

- o Data Safeguards
- o Development of Educational Interventions o Discussion
- o Ethics and Dissemination Specific comments/edits:

Specific comments/edits:

Throughout manuscript:

- In some places “interventions” are mentioned, in others “educational interventions”.
- o Will interventions be limited to educational interventions or might other interventions also be developed? Please clarify.

The interventions will mainly be educational and facilitating implantation of guidelines, we have made this clearer in the manuscript.

- Strengths and Limitations (bullets):
- o Specify that the points listed are strengths/novel contributions.

Updated as suggested.

- Introduction
- o Paragraph 3, sentence 2—Please provide the original citations(s) for work(s) showing that outcomes varied by surgeon.
- o Paragraph 3, sentence 3—Consider moving the citation for Wiebe et al 2002 to the end of sentence 3.
- o Paragraph 3, sentence 6—The sentence states that the implementation of programs was associated with improvements in 7 different outcomes. Were there improvements in all 7 outcomes for each of the programs/initiatives cited? If not, use the citations to indicate which program was associated with improvements in each specific outcome.
- o Paragraph 4, sentence 2—This sentence states “although outcomes have improved”. What outcomes are being referred to? The term “outcomes” is very broad—are these surgical outcomes, long-term outcomes such as overall or disease-free survival? Please be explicit and support with references where possible.
- o Paragraph 5, sentence 4—Consider “examine practice” rather than “quantify practice”. Also, change from “agree areas” to “agree on areas”.
- o Paragraph 6, sentence 1—Should “YCRBCIP” be “YCR BCIP” as it appears earlier on?
- o Paragraph 6, #2—This is the first time Humber is mentioned. Ensure consistency of use throughout.

All Updated as suggested.

- Methods and Analysis
- o Data Collection
- It would be helpful if information on all of the individual data sources were addressed under the heading of Data Collection.

This is included in the revised layout.

- 2(c) is “Screening for Lynch syndrome and deficient mismatch repair on routinely diagnosed bowel cancers”. How is this a data source? Will all of these individuals be screened as part of the study? Or is screening data being collected from elsewhere? Please clarify.

We have removed this as a ‘data source’, as we can see why this might be confusing. This falls under facilitating implementation of guidelines and is perhaps better suited to a separate manuscript.

- Will there be a patient-level linkage between the various data sources? If so, please provide details (preferably in a separate section focusing entirely on linkage). We have added a new section for this.

o Patient Recruitment

- Who exactly will be approaching patients and inviting them to participate/have their data included in the study? If a member of patient’s clinical care team, this raises issues around coercion. How will this be addressed?

The patients will be identified and recruited via the clinical care team with the Participant Information Sheet explained to patients by specialist oncology research nurses or Clinical Nurse Specialist (CNS). This is standard clinical trial/study practice in the UK as it is only the clinical team who should have knowledge of the patients prior to patient consent. This being the case, all those undertaking the consent process have undertaken Good Clinical Practice (GCP) training. GCP is the international ethical, scientific and practical standard to which all clinical research is conducted. Compliance with GCP provides public assurance that the rights, safety and wellbeing of research participants are protected (<https://www.nihr.ac.uk/our-research-community/clinical-research-staff/learning-and-development/national-directory/good-clinical-practice/>). All staff undertaking patient consent must explain to potential participants that their agreement to participate must be voluntary and free from coercion or undue influence. In addition, the research ethics committee have reviewed the study documentation and protocol, and accepted the procedures for consenting participants meet the ethical standards required (see page 15). We have added to the manuscript on Page 8.

o Patient Participation

The majority of content in this section is related to PROMs and could be described under a description of PROMs (ideally under "Data Collection").

This is included in the revised layout.

o Analysis

I suggest that the authors list and define all individual variables that will be included in the study, and identify the relevant data source(s) for each. This can be done in a table.

The manuscript details various data sources, but they do not all appear in the analysis section. As such, it is unclear how all of the data sources will be used. This section would benefit from additional information. For example, what are the specific research questions that the authors plan to address first? Are there specific planned sub-studies? Will there be a multi-phased approach?

We have revised the analysis section and added a Table to explain the measures being analysed and the various sources these come from for better clarity.

o References

#4 and #5 refer to the same article. Please address error and update list accordingly.

Updated as suggested.

Reviewer: 2

Reviewer Name: Paul Ritvo PhD

Institution and Country: York University, Toronto, Canada

Please state any competing interests or state 'None declared': None declared

Please leave your comments for the authors below

This protocol paper write-up is deceptively simple as the project has required a great deal of multidisciplinary coordination to come about. I can appreciate the emphasis on streamlining the complexities involved but I believe the authors have gone too far in this regard. I would like to read more background on how the necessary coordination was accomplished. I also think that more detail is required on the 'routinely collected clinical datasets' re: what exactly they are and how they are routinely collected. With respect to the 'new information on aspects of care not currently quantifiable through

existing datasets' how are these new data going to be collected, i.e. what deviations from standard procedures are being enacted and how? We need to know these additional details to evaluate quality and to possibly emulate the effort in other jurisdictions.

We thank the review for the comments and have revised the manuscript accordingly.

The study is a complex one and difficult to fit into a standard protocol description which is why we have not expanded the detail in some sections. We have now tried to balance the explanation and detail by expanding some sections.

The coordination of the study is important and we have emphasized this by including it as a limitation. The success of the study relies on engagement from MDTs and clinicians; we have updated the study overview and figure to try and give more details of this and how it will be achieved.

We have included a table for the routinely collected clinical datasets and what variables/outcomes will be used from them to give more detail on this.

We have restructured the methods to make it clearer and flow better. We have broken down each section into 3 parts - Existing population-based datasets, Direct PROMs and tissue collection, Audit and survey data. We have given more detail for each, although the 'Audit and survey data' is more fluid than the other two data sources as the exact nature of these are to be discussed and agreed upon by each clinical discipline. These shall form individual publications in the future with specific methods described in detail at that point.

VERSION 2 – REVIEW

REVIEWER	Cynthia Kendell Research Associate, Cancer Outcomes Research Program, Nova Scotia Health Authority/Dalhousie University, Canada
REVIEW RETURNED	16-Aug-2019

GENERAL COMMENTS	REVIEWER FEEDBACK Thank you for the opportunity to review the revised manuscript. The work being undertaken by the authors is complex, requiring multiple data sources and a multi-phased approach. The challenge to the authors is to carefully consider how to present it in a way that is clear and offers an appropriate level of detail to allow the reader to assess the quality of the work, and to provide guidance for other researchers who may embark on a similar program of research. Substantial revisions are required to achieve this. Please see comments below. General comments/recommendations:  o Currently, the content does not flow well. Headings were suggested to the authors to help improve organization and flow of the document. Although these headings were incorporated, the text was not revised accordingly. As a result, the content under each heading does not always “fit” the heading. Some reorganization of the text is required if the authors opt to use the suggested headings. o The presentation of information in the manuscript requires the reader to jump ahead in order to get the information necessary to understand sections that appear early on. Substantial work is required to present the content in a way that flows logically and improves readability (with or without use of the suggested headings). Specific comments/recommendations (by section/subsection):  o Abstract  [ ] Regarding the text, “a region representative of the nation in these terms”, what are the terms being referred to? o Strengths and Limitations  [ ] Bullet #1 and #2 are incomplete sentences and would benefit from rewording, similar to that used in #3. o Introduction
--

	 □ Paragraph 2, sentence 1. The use of the term “variability” is questionable. Undoubtedly, variations in care exist in the UK, but this is true in all countries, so it is not clear why variability is believed to be a particular issue in the UK and thought to contribute to poorer survival compared to other countries. □ Paragraph 4, sentence 1. The authors state that “Whilst these programmes have undoubtedly had a positive impact, variations in management remain and particularly so in the UK.” Please include references to support the assertion that variations in management in the UK are an issue. □ Paragraph 4, sentence 2. Revise sentence to state, “Although survival rates in the UK have improved, they remain lower than in comparable countries [10]”. □ Paragraph 4, sentence 3. What are the variations in care that are being referred to here? And why are they thought to contribute to poorer survival? □ Paragraph 5. Include a reference for YCR. Add quotations around the “avoid, survive, and cope with cancer” as this came directly from the YCR website. □ Paragraph 5, sentence 3. There is mention of “several” English National programs, but only 2 are cited. □ Paragraph 5, sentence 4. Change “and agree areas for improvement” to “and agree on areas for improvement” □ Paragraph 5. Change “implemented” to “developed and implemented”. □ Paragraph 6. For consistency, each item in the list should begin with a verb, and the same verb tense used throughout. For example, #2 should begin with “determine” and #3 with “provide”. □ It may be helpful if the objective of the paper was provided. Clearly, this is a protocol paper, but an explicit statement of the objective would help set the stage for the reader. o Methods and Analysis  □ This section requires substantial revisions throughout to improve clarity. New headings were added, but the content was not revised accordingly. Reorganization of content is required to ensure the information is presented in an order that is logical for the reader, and that minimizes duplication of information. □ Throughout this section, audit and survey data are not well explained. Additional information is required. For example: Who exactly is being audited? What performance measure will be used and how will they be identified? What is the objective of the survey? What topics will be addressed? □ Programme Overview  • Paragraph 1. Replace “interrogate” with “examine” or “analyze”. • Paragraph 1. Change “The programme will run from 1st April 2016 until 31 March 2021” to “The programme commenced on 1st April 2016 and will run until 31 March 2021” seeing as the program is already in progress. • Paragraph 2, sentence 4. Replace “an individual established from each” with “an individual identified from each” or “selected from each”. □ Study Cohort/Participants  • The content under this heading does not actually correspond to the heading. The content that is provided here is about data sources, not the individuals that will be studied. The authors may wish to create a “Data Sources” section. • If the authors include a “Study Cohort/Participants” section, the section should address who is being studied, and set out the inclusion and exclusion criteria for each data source.
--	---

	Alternatively, this information could be incorporated into the “Cohort Identification and Patient Recruitment” section.  [ ] Cohort Identification and Patient Recruitment  • Regarding the statement “Eligible patients will be identified and approached in two ways”, what are the eligibility criteria for individuals complete PROMs and providing tissue samples? • How will MDTs be invited to participate in the survey? [ ] Consent  • Paragraph 1. With regard to existing population-based datasets, the NCRAS may be legally permitted to collect data for purposes directly related to care delivery, but this does not necessarily mean it can be disclosed to a research team without individual consent. Often there are legislative and ethical provisions to permit the disclosure of information for research purposes without individual consent under certain conditions. Please clarify. • Paragraph 2. Under “Direct PROMs and tissue collection”) replace hyperlink with a numbered citation. • Will individuals be able to consent to only PROMs or only tissue sample, or must they consent to both? Please clarify. • Will the authors rely on implied consent for the survey? [ ] Data Collection  • The information provided here is very helpful and should appear earlier on in the paper (e.g., under “Data Sources”, if the authors choose to keep that heading/section). • Paragraph 2. This is the first time that it is mentioned that patients will be asked to consent while in hospital. This should be mentioned earlier on when consent is addressed. [ ] Data Linkage  • How will this linkage occur? For example, via encrypted healthcard numbers, unique study identifiers, etc. [ ] Data Analysis  • Under “Direct PROMs and tissue collection”, paragraphs 1, 2, and 4 are not about analysis. This content would fit better elsewhere (i.e., data sources, data collection, and participant recruitment). • Under “Direct PROMs and tissue collection”, the last two paragraphs are related to the tissue samples. The content describes the analysis that the tissue will be subjected to, but does not explain how that information will subsequently be used by the research team. • See prior comment regarding “Audit and survey data”. • Table 1 is very helpful. [ ] Data Safeguards  • One of the major privacy issues related to this type of work is the identification/re-identification of individuals, especially where data linkage is involved. What strategies are in place to prevent identification/re-identification? For example, are direct identifiers being removed from the dataset?  o Ethics and dissemination [ ] According to the journal guidelines, information on “ethical and safety considerations and any dissemination plan (publications, data deposition and curation)” should be included in this section. The importance of privacy protection in this work and data safeguards could be addressed here.  o Figure 2 [ ] Change “Collection of addition data” to “Collection of additional data”, “Agreement of” to “Agreement on”, and “discussion if” to “discussion of”. Also, ensure consistent use of periods.
--	---

VERSION 2 – AUTHOR RESPONSE

Reviewer: 1

Reviewer Name: Cynthia Kendell

Institution and Country: Research Associate, Cancer Outcomes Research Program, Nova Scotia Health Authority/Dalhousie University, Canada

Please state any competing interests or state 'None declared': None declared.

Thank you for the opportunity to review the revised manuscript. The work being undertaken by the authors is complex, requiring multiple data sources and a multi-phased approach. The challenge to the authors is to carefully consider how to present it in a way that is clear and offers an appropriate level of detail to allow the reader to assess the quality of the work, and to provide guidance for other researchers who may embark on a similar program of research. Substantial revisions are required to achieve this. Please see comments below.

We thank the reviewer for the helpful comments and have revised the manuscript accordingly. In particular, we feel the comments on structure have resulted in a better flow of the manuscript.

General comments/recommendations:

- Currently, the content does not flow well. Headings were suggested to the authors to help improve organization and flow of the document. Although these headings were incorporated, the text was not revised accordingly. As a result, the content under each heading does not always “fit” the heading. Some reorganization of the text is required if the authors opt to use the suggested headings.
- The presentation of information in the manuscript requires the reader to jump ahead in order to get the information necessary to understand sections that appear early on. Substantial work is required to present the content in a way that flows logically and improves readability (with or without use of the suggested headings).
- Regarding the two comments above; We have made changes to the structure as suggested and think these make it flow better. It should be noted this is not a simple study and is difficult to fit into a rigid structure that many other studies comply too. However, we feel given the reviewers suggestions and the heading used now fit the text better and make it easier to understand.
- Specific comments/recommendations (by section/subsection):

Abstract

- Regarding the text, “a region representative of the nation in these terms”, what are the terms being referred to?
- Here we mean that colorectal patients in the region are representative of the nation. We have removed ‘in these terms’ as this is implied.

Strengths and Limitations

- Bullet #1 and #2 are incomplete sentences and would benefit from rewording, similar to that used in #3.

- Updated.

Introduction

- Paragraph 2, sentence 1. The use of the term “variability” is questionable. Undoubtedly, variations in care exist in the UK, but this is true in all countries, so it is not clear why variability is believed to be a particular issue in the UK and thought to contribute to poorer survival compared to other countries.
- Excess variability could be a factor but we agree this isn’t clear and have removed the term variability as a specific explanation.
- Paragraph 4, sentence 1. The authors state that “Whilst these programmes have undoubtedly had a positive impact, variations in management remain and particularly so in the UK.” Please include references to support the assertion that variations in management in the UK are an issue.
- Two recent references have been added which show large variation in the management of patients.
- Paragraph 4, sentence 2. Revise sentence to state, “Although survival rates in the UK have improved, they remain lower than in comparable countries [10]”.
- Updated.
- Paragraph 4, sentence 3. What are the variations in care that are being referred to here? And why are they thought to contribute to poorer survival?
- We have given references as an example of this.
- Paragraph 5. Include a reference for YCR. Add quotations around the “avoid, survive, and cope with cancer” as this came directly from the YCR website.
- Updated.
- Paragraph 5, sentence 3. There is mention of “several” English National programs, but only 2 are cited.
- The two we cite are the larger of the programs and the most relevant, we have removed “several” from the sentence.
- Paragraph 5, sentence 4. Change “and agree areas for improvement” to “and agree on areas for improvement” □ □ Paragraph 5. Change “implemented” to “developed and implemented”.
- Updated.
- Paragraph 6. For consistency, each item in the list should begin with a verb, and the same verb tense used throughout. For example, #2 should begin with “determine” and #3 with “provide”.
- Updated.
- It may be helpful if the objective of the paper was provided. Clearly, this is a protocol paper, but an explicit statement of the objective would help set the stage for the reader.
- We agree this would be helpful and have stated this in the last paragraph of the introduction.

Methods and Analysis

- This section requires substantial revisions throughout to improve clarity. New headings were added, but the content was not revised accordingly. Reorganization of content is required to ensure the information is presented in an order than is logical for the reader, and that minimizes duplication of information.
- We have changed the 'cohort/participants' section to 'data sources'. The 'cohort identification and patient recruitment' is now combined and moved with a 'data collection' section.
- Throughout this section, audit and survey data are not well explained. Additional information is required. For example: Who exactly is being audited? What performance measure will be used and how will they be identified? What is the objective of the survey? What topics will be addressed?
- We have made is clearer that this is from clinicians and we give examples; e.g. surgical specimen photographs, scans, surveys. The nature of the study means that the exact data to be collected is not agreed beforehand and is determined by each discipline at regional meetings. We have tried to expand description on the data where relevant.

Programme Overview

- Paragraph 1. Replace "interrogate" with "examine" or "analyze".
- Updated.
- Paragraph 1. Change "The programme will run from 1st April 2016 until 31 March 2021"to ""The programme commenced on 1st April 2016 and will run until 31 March 2021"
- Updated.
- Paragraph 2, sentence 4. Replace "an individual established from each" with "an individual identified from each" or "selected from each".
- Updated.

Study Cohort/Participants

- The content under this heading does not actually correspond to the heading. The content that is provided here is about data sources, not the individuals that will be studied. The authors may wish to create a "Data Sources" section.
- See above.
- If the authors include a "Study Cohort/Participants" section, the section should address who is being studied, and set out the inclusion and exclusion criteria for each data source. Alternatively, this information could be incorporated into the "Cohort Identification and Patient Recruitment" section.
- See above.

Cohort Identification and Patient Recruitment

- Regarding the statement "Eligible patients will be identified and approached in two

ways”, what are the eligibility criteria for individuals complete PROMs and providing tissue samples?

- This may have been a problem the reviewer alludes to with flow of the paper (now rectified) as this is mentioned in the ‘Data Sources’: Every colorectal cancer patient (ICD10 C18-C20) is eligible if they are considered suitable for treatment, English language literate, aged at least 18 years old and with capacity to give informed consent.
- How will MDTs be invited to participate in the survey?
- Invitations will be done through Email and speciality group meetings.

Consent

- Paragraph 1. With regard to existing population-based datasets, the NCRAS may be

legally permitted to collect data for purposes directly related to care delivery, but this does not necessarily mean it can be disclosed to a research team without individual consent. Often there are legislative and ethical provisions to permit the disclosure of information for research purposes without individual consent under certain conditions. Please clarify.

Access to this data is controlled by the Public Health England Office for Data Release. Cancer registration data will only be approved for release where the data is being used for a medical purpose. These permitted medical purposes are described in the Health Service (Control of Patient Information) Regulations 2002 and include: surveillance, clinical audit, service evaluation, ethically approved research, genetic counselling. However, the purpose of this paper is not to describe all health data in detail. We have updated to acknowledge that access must be done through PHE-ODR. We have updated the consent section to include this information.

- Paragraph 2. Under “Direct PROMs and tissue collection”) replace hyperlink with a numbered citation.
- Updated.
- Will individuals be able to consent to only PROMs or only tissue sample, or must they consent to both? Please clarify.
- Patients can only consent to both, we have updated to make this clearer.
- Will the authors rely on implied consent for the survey?
- No individual patient information will be used in audit/survey data.

Data Collection

- The information provided here is very helpful and should appear earlier on in the paper (e.g., under “Data Sources”, if the authors choose to keep that heading/section).

See above.

- Paragraph 2. This is the first time that it is mentioned that patients will be asked to consent while in hospital. This should be mentioned earlier on when consent is addressed.□

The reordering of text makes this clearer with the eligibility and identification of patients mentioned earlier in the data collection section.

Data Linkage

- How will this linkage occur? For example, via encrypted healthcard numbers, unique study identifiers, etc.
- Name, NHS number and date of birth – updated.

Data Analysis

- Under “Direct PROMs and tissue collection”, paragraphs 1, 2, and 4 are not about analysis. This content would fit better elsewhere (i.e., data sources, data collection, and participant recruitment).

See above.

- Under “Direct PROMs and tissue collection”, the last two paragraphs are related to the tissue samples. The content describes the analysis that the tissue will be subjected to, but does not explain how that information will subsequently be used by the research team.
- See prior comment regarding “Audit and survey data”.
- See above.
- Table 1 is very helpful.

Data Safeguards

- One of the major privacy issues related to this type of work is the identification/re-identification of individuals, especially where data linkage is involved. What strategies are in place to prevent identification/re-identification? For example, are direct identifiers being removed from the dataset?
- As the second paragraph explains, data will be pseudonymised with no identifiers leaving the secure environment for analysis: Electronic data with pseudonymised (allocated ID number) patient information will be stored in a secure environment. These files will only be accessible to relevant members of the study analysis team. Where temporary storage of sensitive data is required (e.g. contact details for sending out repeat surveys), files will be accessible only to relevant members of the research team and not stored with any linked data. Members of the analysis team will not have access to any identifiable data.

Ethics and dissemination

- According to the journal guidelines, information on “ethical and safety considerations and any dissemination plan (publications, data deposition and curation)” should be included in this section. The importance of privacy protection in this work and data safeguards could be addressed here.
- The protocol has gone under detailed ethical review and this is stated here.

Figure 2

- Change “Collection of addition data” to “Collection of additional data”, “Agreement of” to

- “Agreement on”, and “discussion if” to “discussion of”. Also, ensure consistent use of periods.
- Updated.

Potential limitations are not addressed.

The success of the study relies on engagement from clinicians in the regional colorectal MDTs and we see this as the main limitation – there is reduced potential to change anything if it is not discussed and agreed with MDT members. We have updated the last paragraph of the discussion and strength/limitation bullet points to reflect this.

VERSION 3 – REVIEW

REVIEWER	Cynthia Kendell Cancer Outcomes Research Program, Dalhousie University/Nova Scotia Health Authority
REVIEW RETURNED	30-Sep-2019

GENERAL COMMENTS	The authors have addressed the majority of my previous comments and made substantial revisions to the manuscript since the original submission. The content has been much improved in terms of clarity and flow. A number of additional comments have been provided below. Title  • The title is improved, however I suggest removing the word “intensive” Abstract  • Introduction, Sentence 3—Replace “instigating” with “developing” or “developing and implementing”. • Ethics and Dissemination, Sentence 1—Replace “seeking” with “aiming” Strengths and limitations  • First bullet—Change “develop” to “developing” Introduction  • Paragraph 4, Sentence 3—Replace “Better understanding” with “A better understanding”. • Paragraph 6, Sentence 1—The font in the first half of the sentence is different from the rest. • Paragraph 6, Aim #3—Need period at the end of the sentence. Methods and Analysis  • Setting  o Change “10% of English diagnoses” to “10% of diagnoses in England”. • Data Sources  o General comments  [ ] The inclusion criteria for the PROMs and tissue collection are provided under data sources. For consistency, this should be done for all three data sources. For
--

	“Existing population-based datasets” the individuals that will be included in the cohort should be specified, as well as relevant inclusion and exclusion criteria. For “Audit and survey data”, will all MDTs be “audited” and invited to participate in a survey/surveys?  o Audit and Survey Data  [ ] Sentence 2— I recommend changing “depending on the needs and availability of the data” to “depending on the informational needs of the MDTs and availability of the data”. [ ] Please provide more information about what kinds of information will be obtained from the clinicians. What are clinicians going to be asked about? I’m sure the authors have a sense of what the survey/surveys might be about so it should be made clear to the reader (a couple of sentences would suffice). • Data Collection  o Existing Population-Based Datasets  [ ] Sentence 1—This is the first time NCRAS appears in the paper so it should be written out. [ ] Sentence 3—The way this sentence is written suggests that the 10 years of baseline data will be updated to assess changes, but I don’t think this is what is meant. Please revise for clarity. [ ] Related to above comment, please clearly define the study timeframe. Data will be obtained from 10 years prior to what (e.g., the start date of the YCR BCIP)? And what is the end of study date? Or data continue to be collected and linked prospectively? [ ] Sentence 4—This sentence states that individuals will be identified and “assigned a managing MDT”. This seems like a very important part of what is being done, but with very little information about how or why. Based on the numbers you provided there are around 30,000 people in England diagnosed with colorectal cancer each year. The data sets will span 10+ years, which means over 300,000 people with colorectal cancer will be identified. How will individuals be assigned to an MDT? How many to each MDT? Will all cases be assigned to MDTs, or just regional? What is the role of the MDT with respect to the cases that are assigned to it? o Direct PROMs and tissue collection  [ ] There is a bit of jumping around in this section that makes it a little hard to follow. It starts with PROMs, moves onto tissue collection, and then goes back to PROMs. I suggest putting all of the PROMs content together and then putting the tissue collection after the PROMs. [ ] Paragraph 6 refers to PROMs and then Paragraph 7 starts with “The survey will be completed”. There should be a sentence at the beginning of paragraph 7 explicitly stating the PROMs will be collected via survey, otherwise the mention of the survey seems abrupt. [ ] Remove the sentence “The same PROMs are included at both timepoints”. The text that follows indicates that there are differences in what will be collected. [ ] Change “the survey is estimated is about 30-35 minutes” to “the survey is estimated to take 30-35 minutes to complete”. A transitional sentence is needed before the survey sections are listed. See suggested wording in next comment. [ ] The way that Sections 4 and Sections 5 of the PROMs are currently described is extremely confusing (“Section Four: The financial cost of cancer (Time 2 only)” and “Section Four: Questions about you (Time 1) and Section Five: Questions about you (Time 2)”). The numbers of the sections within the surveys are not important for the purposes of the manuscript, rather, it is
--	---

	important to clearly show what participants are being asked and when. I suggesting revising as shown below. “The sections that will be included in the survey are: Your overall health and quality of life (both time points) Your everyday life and well-being (both time points) Managing your health (both time points) Questions about you (both time points) The financial cost of cancer (second time point only)”  o Audit and survey data [ ] Sentences 1 and 2—Suggested re-wording: “Audit data will be limited to data that are collected as a part of routine patient care. For example, anonymized MRI scans that have been performed as a part of patient care may be used as part of an educational training initiative. ” [ ] Sentence 4—Move this sentence so that it becomes sentence 3 in the paragraph. By doing this all of the content related to the audit data will be grouped together, and all of the content related to the survey data. [ ] The purpose of the survey/surveys is not clear. Will the survey be asking clinicians about aspects of their clinical practice? Will they be used to assess guideline implementation? Please clarify.  • Consent o Existing population-based datasets [ ] The authors have added content here related to the conditions for data access for NCRAS data, however, a sentence is required that explicitly states that the research team has been granted access to NCRAS for their study (the current sentence states that access can be granted for certain purposes, but not that access has been granted for this particular initiative). [ ] NCRAS is not the only dataset being accessed. What about these other datasets? How are the authors able to access these data without consent? o Direct PROMs and tissue collection [ ] Sentence 7—Add parenthesis around the text “e.g., Lynch syndrome or deficient mismatch repair status” [ ] Sentence 8 —This seems more related to data collection than to consent. Is this something specific that the participants must consent to? o Audit and survey data [ ] The content provided relates only to the audit data, however, clinicians will be invited to participate in a survey/surveys as well. How will consent be addressed for this data collection? If the authors are relying on implied consent (i.e., completion of the survey implies that they have consent to participate), this should be explicitly stated.  • Data Linkage o Sentence 1—The word “together” appears twice.  • Analysis o Existing population-based datasets [ ] Sentence 1— “(ICD10 C18-C20)” can be removed here as it appears earlier on in the manuscript. o Direct PROMs and tissue collection [ ] Tissue collection has been identified as a data source, yet it is unclear how these data will be used from a research perspective (their clinical use is clear). The authors have indicated that these data will be linked to PROMs and existing datasets, so presumably, they are also going to be included in statistical analysis. Please explain.
--	---

	 Table 1—The variables that have been identified under “Audit and survey data” cannot be obtained from survey data. The table should clearly indicate that these variables will come from audit data. This does bring up the question once again, of what types of information that authors hope to obtain from surveying clinicians. If there are specific variables that will be examined, these should also be included in Table 1.
--	---

VERSION 3 – AUTHOR RESPONSE

The authors have addressed the majority of my previous comments and made substantial revisions to the manuscript since the original submission. The content has been much improved in terms of clarity and flow. A number of additional comments have been provided below.

Title

- The title is improved, however I suggest removing the word “intensive”

Abstract

- Introduction, Sentence 3—Replace “instigating” with “developing” or “developing and implementing”.
- Ethics and Dissemination, Sentence 1—Replace “seeking” with “aiming”

Strengths and limitations

- First bullet—Change “develop” to “developing”

Introduction

- Paragraph 4, Sentence 3—Replace “Better understanding” with “A better understanding”.
- Paragraph 6, Sentence 1—The font in the first half of the sentence is different from the rest.
- Paragraph 6, Aim #3—Need period at the end of the sentence.

All the above have been updated as suggested.

Methods and Analysis

- Setting
 - Change “10% of English diagnoses” to “10% of diagnoses in England”.

Updated.

- Data Sources
 - General comments
 - The inclusion criteria for the PROMs and tissue collection are provided under data sources. For consistency, this should be done for all three data sources. For “Existing population-based datasets” the individuals that will be included in the cohort should be specified, as well as relevant inclusion and exclusion criteria. For “Audit and survey data”, will all MDTs be “audited” and invited to participate in a survey/surveys?

We have moved the inclusion criteria from data collection to data sources for consistency and explained that all regional MDTs will be eligible to participate in the audits and surveys.

- Audit and Survey Data
 - Sentence 2— I recommend changing “depending on the needs and availability of the data” to “depending on the informational needs of the MDTs and availability of the data”.

Updated.

- Pleas provide more information about what kinds of information will be obtained from the clinicians. What are clinicians going to be asked about? I’m sure the authors have a sense of what the survey/surveys might be about so it should be made clear to the reader (a couple of sentences would suffice).

We described a few examples of the audits/surveys which are likely to be implemented in the first instance.

- Data Collection

- o Existing Population-Based Datasets

- Sentence 1—This is the first time NCRAS appears in the paper so it should be written out. Updated.

- Sentence 3—The way this sentence is written suggests that the 10 years of baseline data will be updated to assess changes, but I don't think this is what is meant. Please revise for clarity.

- Related to above comment, please clearly define the study timeframe. Data will be obtained from 10 years prior to what (e.g., the start date of the YCR BCIP)? And what is the end of study date? Or data continue to be collected and linked prospectively?

We have reworded this sentence to make it clearer, removing the '10 years' with 01/01/2005 to explain this will form the baseline and data will be collected throughout the programme until the end of the study.

- Sentence 4—This sentence states that individuals will be identified and “assigned a managing MDT”. This seems like a very important part of what is being done, but with very little information about how or why. Based on the numbers you provided there are around 30,000 people in England diagnosed with colorectal cancer each year. The data sets will span 10+ years, which means over 300,000 people with colorectal cancer will be identified. How will individuals be assigned to an MDT? How many to each MDT? Will all cases be assigned to MDTs, or just regional? What is the role of the MDT with respect to the cases that are assigned to it?

We have now explained that hospital admission data will primarily be used to assign MDTs and that the assigned MDT will assumed to have been responsible for the management of the colorectal cancer patient.

- o Direct PROMs and tissue collection

- There is a bit of jumping around in this section that makes it a little hard to follow. It starts with PROMs, moves onto tissue collection, and then goes back to PROMs. I suggest putting all of the PROMs content together and then putting the tissue collection after the PROMs. Updated.

- Paragraph 6 refers to PROMs and then Paragraph 7 starts with “The survey will be completed”. There should be a sentence at the beginning of paragraph 7 explicitly stating the PROMs will be collected via survey, otherwise the mention of the survey seems abrupt.

We have now made earlier mentions of this in both data sources and collections. We also have replaced 'survey' with 'questionnaire' to avoid any confusion with audit/survey data.

- Remove the sentence “The same PROMs are included at both timepoints”. The text that follows indicates that there are differences in what will be collected. Change “the survey is estimated is about 30-35 minutes” to “the survey is estimated to take 30-35 minutes to complete”. A transitional sentence is needed before the survey sections are listed. See suggested wording in next comment. Updated

- The way that Sections 4 and Sections 5 of the PROMs are currently described is extremely confusing (“Section Four: The financial cost of cancer (Time 2 only)” and “Section Four: Questions about you (Time 1) and Section Five: Questions about you (Time 2)”). The numbers of the sections within the surveys are not important for the purposes of the manuscript, rather, it is important to clearly show what participants are being asked and when. I suggesting revising as shown below. “The sections that will be included in the survey are: Your overall health and quality of life (both time points) Your everyday life and well-being (both time points) Managing your health (both time points) Questions about you (both time points) The financial cost of cancer (second time point only)” Updated.

- o Audit and survey data

- Sentences 1 and 2—Suggested re-wording: “Audit data will be limited to data that are collected as a part of routine patient care. For example, anonymized MRI scans that have been performed as a part of patient care may be used as part of an educational training initiative.”

Updated

Sentence 4—Move this sentence so that it becomes sentence 3 in the paragraph. By doing this all of the content related to the audit data will be grouped together, and all of the content related to the survey data.

Updated.

The purpose of the survey/surveys is not clear. Will the survey be asking clinicians about aspects of their clinical practice? Will they be used to assess guideline implementation? Please clarify.

These surveys will be used to assess clinical practice and patient management. We have given an example of a survey regarding the use of adjuvant chemotherapy being used to inform a regional treatment guideline.

- Consent

- o Existing population-based datasets

The authors have added content here related to the conditions for data access for NCRAS data, however, a sentence is required that explicitly states that the research team has been granted access to NCRAS for their study (the current sentence states that access can be granted for certain purposes, but not that access has been granted for this particular initiative).

NCRAS is not the only dataset being accessed. What about these other datasets? How are the authors able to access these data without consent?

The NCRAS data and other datasets are being accessed through the UK Colorectal Cancer Intelligence Hub who have approval from PHE, we have given the specific data sharing agreement as a reference for any reader wanting to look this up.

- o Direct PROMs and tissue collection

Sentence 7—Add parenthesis around the text “e.g., Lynch syndrome or deficient mismatch repair status”

Updated.

Sentence 8 —This seems more related to data collection than to consent. Is this something specific that the participants must consent to?

This is not a specific consent item, but this seemed an appropriate place to mention this.

- o Audit and survey data

The content provided relates only to the audit data, however, clinicians will be invited to participate in a survey/surveys as well. How will consent be addressed for this data collection? If the authors are relying on implied consent (i.e., completion of the survey implies that they have consent to participate), this should be explicitly stated.

Implied consent has been explicitly stated here now.

- Data Linkage

- o Sentence 1—The word “together” appears twice.

Updated

- Analysis

- o Existing population-based datasets

Sentence 1—“(ICD10 C18-C20)” can be removed here as it appears earlier on in the manuscript.

Updated.

- Direct PROMs and tissue collection

- o Tissue collection has been identified as a data source, yet it is unclear how these data will be used from a research perspective (their clinical use is clear). The authors have indicated that these data will be linked to PROMs and existing datasets, so presumably, they are also going to be included in statistical analysis. Please explain.

Following how the molecular testing is done we have now explained that these results will be linked PROMs and population data and be including in regression analyses to see how tumour biology influences outcomes, response and quality of life.

o Table 1—The variables that have been identified under “Audit and survey data” cannot be obtained from survey data. The table should clearly indicate that these variables will come from audit data. This does bring up the question once again, of what types of information that authors hope to obtain from surveying clinicians. If there are specific variables that will be examined, these should also be included in Table 1.

We have updated to state audit or survey in the table, with text in response to the earlier comment now explaining and giving more examples of this.